# Tribological Research of Resin Composites with the Fillers of Glass Powder and Micro-Bubbles

**DOI:** 10.3390/ma17153764

**Published:** 2024-07-31

**Authors:** Juozas Padgurskas, Vitoldas Vilčinskas, Muhammad Ibnu Rashyid, Muhammad Akhsin Muflikhun, Raimundas Rukuiža, Aušra Selskienė

**Affiliations:** 1Department of Mechanical, Energy and Biotechnology Engineering, Vytautas Magnus University, Akademija, LT-53361 Kauno r, Lithuania; vitoldas.vilcinskas@vdu.lt; 2Department of Mechanical and Industrial Engineering, Faculty of Engineering, Gadjah Mada University, Jln. Grafika No.2, Yogyakarta 55281, Indonesia; muhammadibnurashyid@mail.ugm.ac.id (M.I.R.); akhsin.muflikhun@ugm.ac.id (M.A.M.); 3Department of Characterisation of Materials Structure, State Research Institute Center for Physical Sciences and Technology, Savanorių Ave. 231, LT-02300 Vilnius, Lithuania; ausra.selskiene@ftmc.lt

**Keywords:** resin composite, glass powder fillers, micro-bubbles, friction coefficient, wear, surface analysis

## Abstract

This study investigates the tribological properties of resin composites reinforced with the fillers of glass powder and micro-bubbles. Resin composites were prepared with varying concentrations from 1% to 5% wt of fillers. Tribological tests were conducted using a block-on-ring scheme under dry friction conditions. The measurements of friction coefficient and wear values were performed under variable rotation speeds and loading conditions. The study showed that resin composites with 2–3% glass powder fillers and resin composites with 3–4% micro-bubbles exhibited optimal tribological properties. The resin glass powder modifications reduce the wear by 63% and resin micro-bubbles reduce wear by 32%. SEM analysis of the surfaces revealed surface imperfections and structural damage mechanisms, including abrasive and fatigue wear. The study concludes that specific filler concentrations improve the friction and wear resistance of resin composites, highlighting the importance of material preparation and surface quality in tribological performance. The increased wear resistance on both composites would hopefully expand the usage of additive manufactured composite, namely industrial moving components such as polymer gear, wheel, pulley, etc.

## 1. Introduction

Additive manufacturing (AM), commonly known as 3D printing, is an innovative way to produce components. Unlike subtractive manufacturing, which removes material, AM works by adding material layer by layer. This process offers freedom of geometry, does not require special tooling, allows for rapid production, and results in less material waste [1,2]. These advantages make AM a favorable method in academic research [3], as well as for automotive [4], aerospace [5], medical [6], and electronic components [7]. One of the common techniques of AM is stereolithography (SLA) and direct light processing (DLP). These techniques are known for their higher layer resolution, smoother surface finish, and superior dimensional accuracy [3].

SLA and DLP work by exposing UV light to a photosensitive resin known as vat-photopolymerization resin in a specific pattern with a certain layer height [8]. This UV resin typically produces products with low mechanical characteristics. Unlike material extrusion techniques that require additional processes to combine with other materials such as powder or filler, this method can easily incorporate these materials by directly mixing the powder or filler into the resin, creating a polymer composite [2]. This composite can be used as a lightweight structure, a conductive product for electrical circuits, a nano-composite for MEMS, and insulation for thermal and electrical applications [9,10,11,12].

Polymer composites using filler powders such as CNT, cellulose nanocrystals, and graphene are used to improve mechanical characteristics [13,14,15], which includes also the polymeric coatings of fluoroligomers [16], polymer/silica composites [17], polymer composites with graphene fillers [18], or self-lubricating polymer composite coatings [19]. Several studies have shown the improvement in mechanical strength through the composite powder method. For instance, the usage of 5% wt of graphite increased the compression strength of GFRP–PU composites [20]. Another study demonstrated that adding 4.5% wt of glass powder increased the flexural strength of GFRP composites [21]. Additionally, glass powder has been shown to enhance hardness and reduce wear [22]. Another type of filler, such as air micro-bubbles, can also be implemented in polymer composites. Generally, voids in composites are classified as defects and need to be minimized [23]. However, the use of micro-bubbles can be beneficial for thermal and mechanical properties. It has been proven that the presence of micro-bubbles can increase mechanical strength by up to 70% and reduce specific heat capacity by up to 40% [24].

Tribological studies of modified vat-resin with the fillers show that the process parameters of the manufacturing affect a big proportion of the wear resistance characteristics. It was shown that inadequate or excessive curing leads to increases in the wear rates. Additionally, the layer thickness of specimens also contributes to wear rates [25]. An innovative method to enhance the tribological properties of vat-resin is by combining the resin with cyanate ester oil to create a self-lubricating part. Since the oil and resin have different properties, a certain percentage makes the oil dispersed and trapped inside the cured resin. This leads to reduced wear up to 96% [26]. Contrarily, the drawback of this method was lowering the mechanical properties. The tensile properties were reduced from the first variation. This is caused by the higher concentration of oil, which leads to bigger porosity.

Since the filler is also used as reinforcement for resin, this leads to another idea for improvement in tribological properties. Several studies indicate that using a graphene filler increases tribological performance. A modified vat-resin with MoS_2_ and graphene fillers improved the wear and friction rate by up to 52% and 92%, respectively, under the same lubricant [27]. A study on graphene filler added to vat-resin shows similar results. With only 0.5% wt, the friction coefficient was reduced by up to 50% compared to pure resin [28]. Another study shows that combining oil droplets and mesoporous nano silica reduced the part’s friction coefficient and wear rate by 85.7% and 97.7%, respectively. The presence of the filler compensates for the void from the oil droplets and enhances the tensile and compressive strength [29]. Incorporating ceramic powder was proven to achieve a high-performance vat-resin ceramic composite. From the experiment, the friction coefficient stabilizes around 5000 cycles with a friction coefficient of approximately 0.35. Nevertheless, the material preparations to achieve this product are a bit complex. It required a reagent to suspend the ceramic during the printing process [30].

An investigation was performed into the tribological properties of sealing acrylonitrile–butadiene rubber composite with carbon black functional additives, and it showed wear-reducing and hydrogen exposure effects [31]. The modification of epoxy polymers with Al_2_O_3_ nanoparticles and graphite additives and its influence on the mechanical and tribological properties of the composite were investigated too [32].

However, the tribological properties of vat-resin using glass powder and air micro-bubbles still need to be thoroughly investigated alongside their mechanical properties. The ease of manufacturing these vat-resin composites also warrants exploration. Incorporating glass powder and air micro-bubbles into vat-resin composites can significantly enhance performance, particularly in applications where wear resistance and reduced friction are critical. This study mixed commercially available vat-resin with glass powder and air micro-bubbles to evaluate their effects on tribological properties, such as wear and friction resistance. Understanding the tribological evaluation will expand the use of additive-manufactured resin composites in high-friction resistance environments. This manufacturing technology minimizes the need for molds, allowing vat-resin composites to be rapidly produced in any form for various fields.

This study aims to investigate the possibilities of incorporating glass powder and micro-bubbles modified resin composites for tribological applications in the friction joints of machinery. Compared with the literature review process, this study also gives a new technique to manufacture vat-resin without adding a complex additive to enhance the tribological properties, thus making the manufacturing process much easier.

## 2. Materials and Methods

### 2.1. Materials

This study prepared two kinds of resin composites: resin glass powder (RGP) and resin air micro-bubbles (RMBs). Figure 1 shows the manufacturing process of the composite sample. The resin was obtained from Anycubic Basic Colored UV Sensitive Resin (Shenzhen Anycubic Technology Co., Ltd., Shenzhen, China), and the specification can be seen in Table 1. The shape composite sample dimensions can be seen in Figure 2. Glass powder and air micro-bubbles fillers were sourced from an Indonesian company (PT.Justus Kimia Raya, Semarang, Central Java, Indonesia). The specification of the glass powder can be seen in Table 2. The air micro-bubble average diameter is 50–100 μm. There was no other substance inside the air micro-bubble. The vat-resin was mixed with the powder fillers in varying concentrations from 1% to 5% wt, using a planetary mixer set at 1500 rpm for 5 min. The vat-resin composites with the fillers were then manufactured using an Anycubic Photon Mono SE printer (Shenzhen Anycubic Technology Co., Ltd., Shenzhen, China) and then washed and cured in a UV curing machine for 15 min to obtain fully clean and cured specimens to achieve the optimum polymerization. The printing parameters can be seen in Table 3. The number of samples tested for every specimen was 3.

### 2.2. Methods

The composite samples (18 mm × 10 mm × 6 mm) and ring counter-body samples (∅35 mm × 10 mm) made of steel C45 (LST-EN 10083-1) were prepared for the investigation (Figure 1).

The tribological research was performed using a modernized SMC-2 tribometer (Vytautas Magnus University, Lithuania), based on block-on-ring ASTM G77-17 standard [33] recommendations. The testing scheme is presented in Figure 2.

Tribological research consists of friction coefficient and wear measurements under variable rotational speed regimes, variable load regimes, constant parameter regimes, and multicycle scratch testing. Several block-on-ring tests with variations in tribological parameters were performed.

The variable rotation speed regime shows how materials react to different rotation speeds in dry sliding conditions. A 50 N load and a rotation speed increase of 200 rpm to 900 rpm (with increments of 100 rpm) regime is used. Having 2 steps for lowering the speed after peak rpm is reached, the total sliding distance was 1 km. With each composite type and filler concentration, tests are repeated 3 times.

A variable load regime is created to evaluate how composites react to different loading in dry sliding conditions. A 500 rpm of rotation speed and load increase from 40 N to 100 N with load increments of 20 N was used in this program. With each step, 500 m of dry sliding was performed. The total sliding distance was 2 km.

The friction torque was measured during the test and recalculated into the friction coefficient. The wear of composite samples was evaluated according to the loss of worn mass after the variable speed and variable load tests. Samples weighted before and after each test were determined by electronic balance ABJ 120-4M (Kern & Sohn GmbH, Balingen, Germany) with an accuracy of 0.1 mg. Random-picked samples were double-checked with an optical profilometer for conformity.

Surface investigation (including SEM pictures and EDX) of the samples was performed with a scanning electron microscope (SEM) Helios NanoLab 650 (FEI, Hillsboro, OR, USA). The multicycle scratch tests were conducted to see how each composite type reacts to a sliding steel ball (Ø3 mm) for 1000 cycles with a constant load increase from 0 to 30 N, 9 mm of scratch length, and 300 mm/min speed. The load was applied in both directions—backward and forward. Scratch testing was conducted with micro-combi scratch tester CSM (CSM, Peseux, Switzerland) (Figure 3).

## 3. Results

The tribological properties of the selected materials were analyzed when changing the rotation speed and loading according to the friction coefficient and wear in dry sliding conditions. Besides that, the relation of tribological parameters with the surface investigation was analyzed according to the surface damage, structural and crack analysis, and scratch testing.

### 3.1. Friction Coefficient

The friction coefficient measurements were performed by changing the rotation speed at constant loading, and evaluating constant rotation speed at changeable loading. The constant loading was at 50 N with a rotation speed starting from 200 rpm to 900 rpm, with an increment step of 100 rpm.

Each step passes a 100 m run of dry sliding. Having two steps for lowering the speed after peak rpm is reached, the total sliding distance is 1 km. Figure 4 presents the selected typical curves of three times repeated measurements of friction coefficient variation at changing rotation speeds for RGP and RMB materials.

At the beginning of the tests at a lower speed, the lowest friction coefficient results for RGP composite were achieved when using 0% and 1% glass powder concentration. However, later at higher rotation speed it increases, and at 700–900 rpm, the results of the friction coefficient became similar to all samples (about 0.11), and just the samples of 3% and 4% concentration showed a slightly lower friction at the end. The average friction coefficient of RMB samples was lower compared to RGP samples. The highest friction coefficient was reached with the 1% micro-bubbles concentration sample—0.11—while the lowest was 0.05–0.06 (2%, 4%). Generally, it could be stated that the rotation speed influences the stability of the friction coefficient, but its value does not depend significantly on the speed.

To evaluate how composites react to different loads in dry sliding conditions, a rotational speed of 500 rpm and load increment from 40 N to 100 N (step 20 N) regime were experimented. With each step, a 500 m run of dry sliding was performed. The total sliding distance was 2 km. The measurement results are presented in Figure 5.

Measurements at variable load regime showed that the lowest values of friction coefficient using the RGP composite were obtained on 2% glass powder concentration samples; all other samples demonstrated results similar to each other. The results of the friction coefficient of RMB samples revealed that the lowest friction coefficient had a 4% concentration composite, and the slightly lower friction coefficient had 3% and 2% samples. The 1% sample acquired the highest values (about 0.05 higher), but besides that it had a very unstable curve at the end of the test, reaching a 0.18 friction coefficient.

### 3.2. Wear Resistance

The wear resistance plays an important role when evaluating the longevity of materials to ensure the secure operation of friction pairs in machinery operation. The wear resistance in our study was evaluated according to the loss of worn mass of the samples.

The results of wear measurement for RGP and RMB samples with the dry sliding regimen are presented in Figure 6.

From the mass loss results of the RGP samples at the changing rotation speed, it can be seen that there was a decreasing mass loss tendency as the filler concentration increased up to 2%, and then it can be seen that constant mass loss regularly increased up until the sample of 5% concentration. According to this, the 2% and 3% RGP samples were the best of the rest of the RGP concentrations at variable speed test regimes. In fact, the 2% RGP sample had 3.5 times lower mass loss than the 0% reference sample. The worst result at changeable speed was of the 5% RGP sample, which had almost equal wear as the reference sample.

Adding micro-bubbles to resin (RMB) was less efficient than RGP composite if the concentration of micro-bubbles was lower (1–3%). However, if micro-bubbles concentration reached 4–5%, the wear of RGP samples was lower compared to high-concentration glass powder composites. Comparing the average wear-reducing efficiency of modified samples with all concentrations, it was seen that all RGP modifications reduced the wear by 63%—and RMB by 32%.

When analyzing the wear at the tests with increasing loading, it should be considered that the wear run of such samples was two times longer than at varying speed tests, i.e., wear distance at changeable speed tests was 1000 km, and at varying load tests it was 2000 km. At the testing of RGP composites on changeable load, the mass loss was the lowest in the 1%, 2%, and 3% RGP samples, and was significantly lower than the reference sample results. At a glass powder concentration of 3%, the RGP increased the wear resistance ability almost two times. The results of the 4% and 5% samples were similar to the reference (0%) sample.

Testing the RMB composites, the optimal concentrations of micro-bubbles seem to be 2%, 3%, and 4%, and those RMB samples showed lower mass loss results than the 0% sample. The lowest of all results is obtained by 3%, showing 60% lower wear compared to the reference sample. The worst result was shown in the 5% samples when the mass loss was 66% higher compared to the reference sample.

Considering the average wear loss results and longer wear runs at varying load tests, it could be assumed that RGP and RMB samples have optimal concentrations of the fillers. For both composites, such concentrations are 2% and 3%. It correlates to the results of friction torque measurements of the different composite samples.

### 3.3. SEM Analysis of Composite Surface

The structural analysis of the worn surface of the samples allows us to clarify the reasons for the different tribological properties of the composites.

SEM surface analysis was performed on the worn surface after a 1 km run at 200 rpm and 60 N loading, at dry block-on-ring conditions for the RGP 2% sample (Figure 7) and for the RMB 4% sample (Figure 8). Such concentrations of the composite fillers present the samples that have higher wear resistance and even those samples can show the cause of the surface decay.

Figure 7 shows that the 2% RGP sample exhibits a non-uniform surface structure, whereas glass powder particles display cyclic distribution on the surface (Figure 7a,b). Besides that, on the surface are clearly seen cracks which can lead to the fracture of surface particles. Some of them (Figure 7c,d,f) form the zones of the possible particle breakaway. They show the signs of not only abrasive wear because of the harder steel counter-body, but also the important role of fatigue wear.

The SEM pictures of the RMB sample show the distribution of the micro-bubbles (Figure 8a) and their contact with the surface fracture (Figure 8e,f). Surface damage here has a slightly different character, but the cracking of the surface is also clearly seen (Figure 8a,b). Differently from the RGP sample at the RMB surface, some role can also play in the delamination of the surface particles (Figure 8c,d).

EDX analysis of the RMB sample was performed to clear possible changes in the material content during the friction and wear process at the sample testing. Figure 9 presents the elementary content of the surface material before and after the test.

The EDX pictures show that the content of the main elements of the resin material before and after the tests do not differ significantly. However, after the tests, some signs of iron appear in the contact zone. The presence of iron could be transferred from the steel counter body during the dry friction contact test.

It shows that no chemical transformations are taking place during the tests, only mechanical wear presence.

### 3.4. Multicycle Scratch Testing

The scratch test allows for evaluating the resistance of the material surface to increasing contact loading. One thousand cycle scratch tests have been completed to evaluate how each of the selected composite topography reacts to loading with a sliding steel ball (Ø3 mm) of the intender with constant load increment from 0 to 30 N, 9 mm of scratch length, and 300 mm/min speed.

The scratch depth at RGP and RMB composites at the beginning of the test and after 1000 cycles is presented in Figure 10.

We can see that significantly smaller scratches remain on RGP composites rather than on RMB, both during the first and last cycle. The lowest (1%) and highest (5%) filler concentrations reduce the scratch resistance of the composites, resulting in deeper scratches. The same results are on both RGP and RMB samples, and that trend, which regards concentration and scratch depth, persists during all 1000 cycles. The highest resistance on scratch test presented the RGP samples of 3% and 4% of concentrations and RMB samples of 2% and 3% of concentrations. It passes the results of wear tests, which show the lowest wear of those optimal concentrations. The RGP sample of 3% concentration after 1000 cycles was scratched only for 12.8 µm in the deepest part. It was 60% less than the control version of the composite. The 3% RMB sample was scratched only for 11.7 µm showing 75% lower scratch depth compared to the control version.

Comparing the scratch resistance results we can see that scratch depth after 1000 cycles on both RGP and RMB composites increases by about 50–70% compared to the beginning of tests. This increase is higher on RMB composites, especially at optimal concentrations (2% and 3%).

It is interesting that at optimal composite concentrations, the scratch is also more stable, and at 1% and 5% concentrations and the reference version sample the scratch path had considerable fluctuations in the beginning and at the end of testing. It shows the non-homogenous structure of those composites especially at lower surface depths.

The surface damage at the scratch testing is presented in SEM pictures of the scratch path (see Figure 11).

The character of surface damage after the scratch testing correlates with the surface fragmentation during the wear tests. When scratching the RGP sample, the clear cracks form on the edges (Figure 11a,c) and in the center of the scratch path (Figure 11b). Magnification of the scratch center (Figure 11d) shows the sandwich-type structure of the affected surface. The pictures of the scratched RMB sample (Figure 11e,f) show a more uniform structure of scratch path surface (Figure 11e) and the magnification of surface fracture (Figure 11f) displays the impact of micro-bubbles on the formation of surface cracks.

## 4. Discussion

The results show that there is some optimal concentration of resin fillers for the tribological properties of the investigated composites. From the measurement of friction losses and wear resistance, we can consider that for RGP composites, such concentration of glass powder additives is about 2–3%, and for RMB composites the concentration of micro-bubbles should be 3–4%.

Such optimal concentrations are not asserted at the beginning of tribological testing; they show their tribological efficiency at higher speed and higher loading (see friction results in Figure 4 and Figure 5). Wear tests confirm that the fluctuating friction in the beginning plays a less important role, and the stable and lower friction coefficient of the samples with optimal concentration at the end is decisive for their higher wear resistance. Summarizing the tribological testing results, we analyzed that the friction coefficient of all tested composite samples depends mainly on the loading, but not on the speed change. The surface wear resistance depends on the structure of the composite reflecting the filler concentration, and the dependence of wear mechanisms (such as abrasive, fatigue, or adhesive) on the regimens of the tests is not observed. The difference in wear values was caused by the intensity of surface decay, not by the different wear mechanisms.

The surface analysis shows the multiple wear characteristics of the resin composites. In opposite to metallic surfaces where the abrasive wear is usually dominating in resin composites, the fatigue wear (it shows the surface cracks and delaminating parts; see Figure 7 and Figure 8) and adhesive wear (it shows some transfer of counter-body iron on the surface of composite; Figure 9) play also an important role. Nevertheless, the wear mechanism analysis of the surface dry friction could change drastically under lubrication condition.

This research concentrated on investigating the possibility of increasing the wear resistance and considering the influence of a lower friction coefficient in dry friction pairs. However, a considerably lower friction coefficient can be achieved when at least limitary lubrication is applied in friction pairs. Our preliminary tests show that adding a very small amount of lubricant decreases the friction coefficient of such composites several times. The increased wear resistance on both composites will hopefully expand the usage of additive manufactured composite, namely industrial moving components such as polymer gear, wheel, pulley, etc. Besides that, the operational use of such composite friction pairs could be oriented to joints where a high friction coefficient and low wear are required (ex., brakes, clutches, etc.).

It means that our study could be considered as a primary investigation of the selected composites. For the decision of possible use of such composites in machinery applications, there should be investigations made on longer operation periods; selected versions of composites should be tested at the lubrication conditions; and other selected composites with high friction coefficients should be tested for wear resistance at dry friction taking into consideration its use in high-friction joints.

## 5. Conclusions

Tribological investigations of successfully manufactured resin composites with glass powder and micro-bubbles show that there are optimal concentrations of the fillers for their tribological properties. Several key findings from this experiment are as follows:The most wear-resistant RGP composites are those with a 2–3% concentration of glass powder (the wear after 2 km run at changeable load is up to two times lower than reference sample) and RMB composites with a 3–4% micro-bubbles concentration (it reduces the wear up to 60%).The value of the friction coefficient of tested samples does not depend significantly on the speed; it is mainly determined by the load (by gradually increasing the load from 40 N to 100 N, the coefficient increases from 0.02–0.06 to 0.14–0.18). The stable and lower friction coefficient is the factor that determines the highest wear resistance.SEM analysis of the worn surface revealed its damage character. During the operation of the samples, scratches and cracks formed on the surface. Abrasive, fatigue, and adhesion wear mechanisms were observed, with glass powder particles leading to non-uniform surface structures and microbubble composites showing delamination effects.The 3–4% concentration RGP composites and 2–3% RMB composites increased the indentation resistance of the composites when the surface was scratched in the range of high loads (15–30 N). This property correlates with the wear resistance of investigated composites.

Based on the conclusion, the vat-resin composite shows an increase in wear resistance by adding both types of filler. This finding expands the possibilities of vat-resin usage in working areas that need constant surface contact. Nevertheless, the surface morphology analysis still needs improvement to achieve better surface structures. Perhaps the texturing method can be implemented into the vat-resin composite in future works.

## Figures and Tables

**Figure 1 materials-17-03764-f001:**
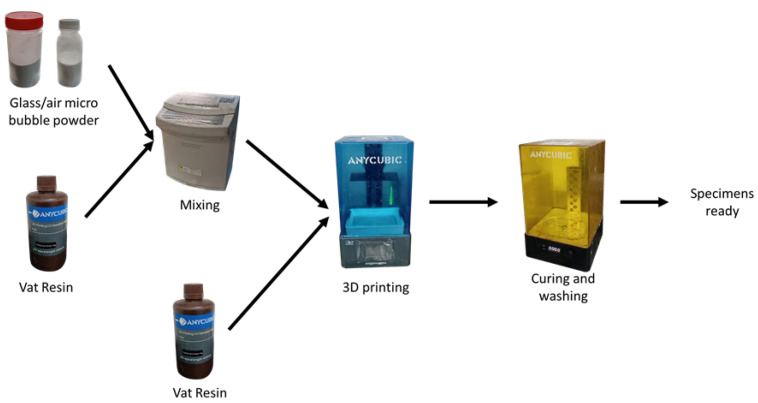
The manufacturing process of the vat-resin composite.

**Figure 2 materials-17-03764-f002:**
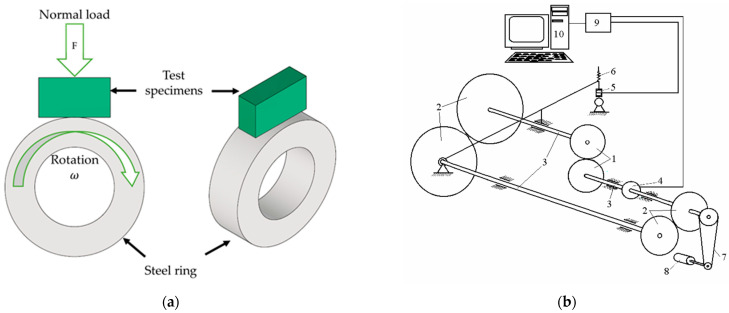
Tribological testing: (**a**) Tribo-pair positioning; (**b**) scheme of tribotester SMC-2: 1—roller; 2—gears; 3—shaft of friction rig; 4—sensor of friction torque; 5—loading sensor ZF-500; 6—loading screw; 7—drive belt; 8—electromotor; 9—registration panel ADC-200/20; 10—computer.

**Figure 3 materials-17-03764-f003:**
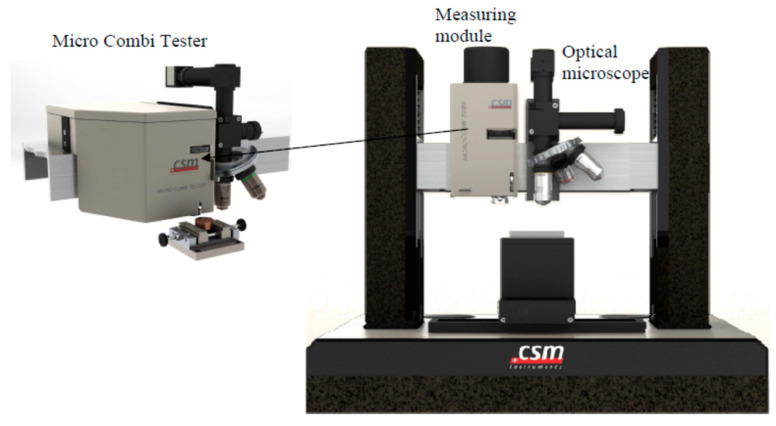
Micro-combi scratch tester CSM.

**Figure 4 materials-17-03764-f004:**
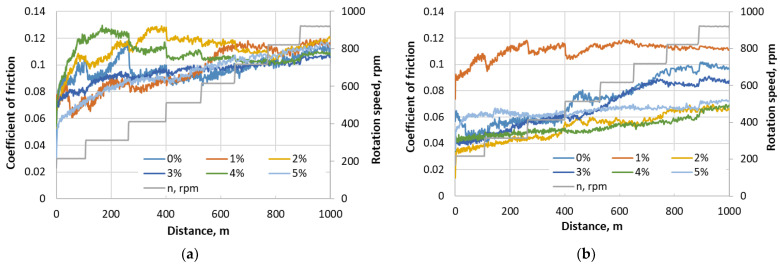
The friction measurement results at constant loading of 50 N at changing rotation speed for different filler concentrations of resin composite: (**a**) with glass powder; (**b**) with micro-bubbles.

**Figure 5 materials-17-03764-f005:**
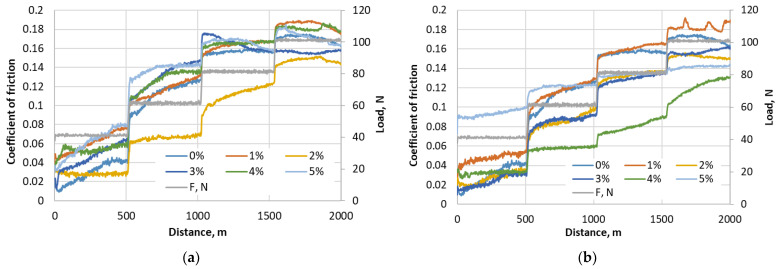
Friction measurement results at a constant rotation speed of 500 rpm at changing loading for different filler concentrations of resin composite: (**a**) with glass powder; and (**b**) with micro-bubbles.

**Figure 6 materials-17-03764-f006:**
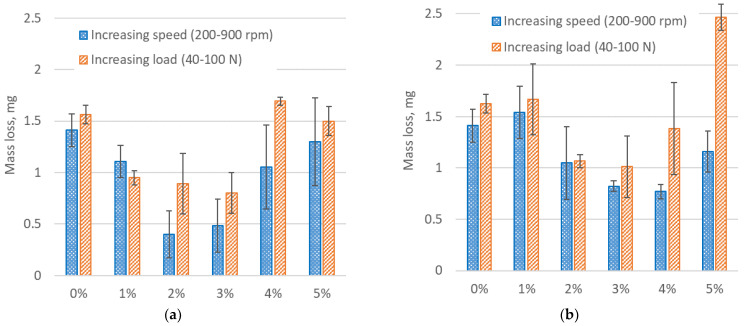
Wear measurement results when changing the rotation speed and loading for different filler concentrations of resin composite: (**a**) with glass powder; and (**b**) with micro-bubbles.

**Figure 7 materials-17-03764-f007:**
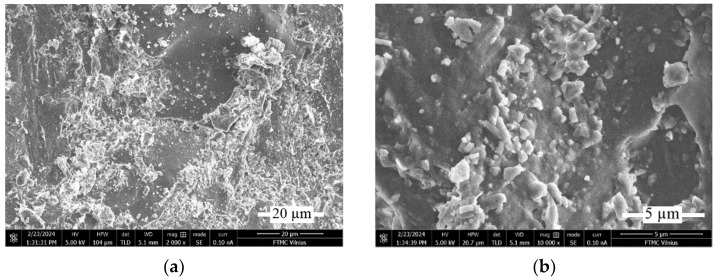
SEM images of the worn surface of 2% concentration RGP sample at different magnification: (**a**) ×2000; (**b**) ×10,000; (**c**) ×1000; (**d**) ×2000; (**e**) ×10,000; (**f**) ×2000.

**Figure 8 materials-17-03764-f008:**
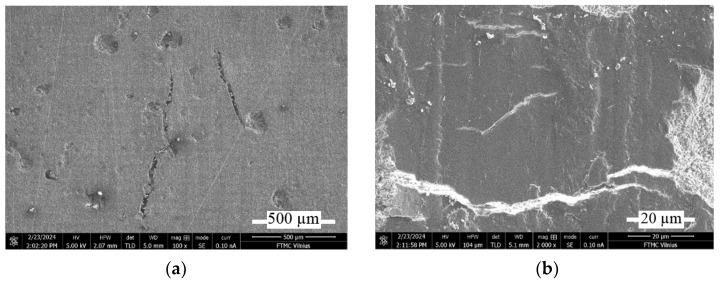
SEM images of the worn surface of 4% concentration RMB sample at different magnification: (**a**) ×100; (**b**) ×2000; (**c**) ×2000; (**d**) ×10,000; (**e**) ×10,000; (**f**) ×2000.

**Figure 9 materials-17-03764-f009:**
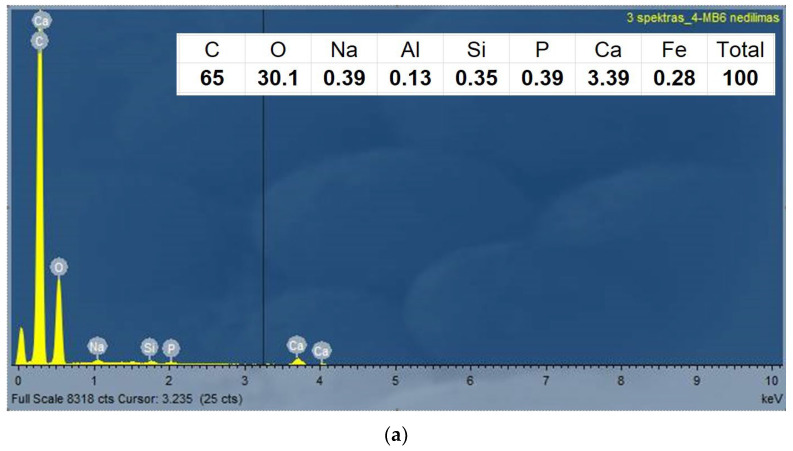
EDX surface analysis of the RMB sample of 4% concentration: (**a**) non-abraded surface; (**b**) the worn zone after the test. The elemental percentage does not present the quantitative values.

**Figure 10 materials-17-03764-f010:**
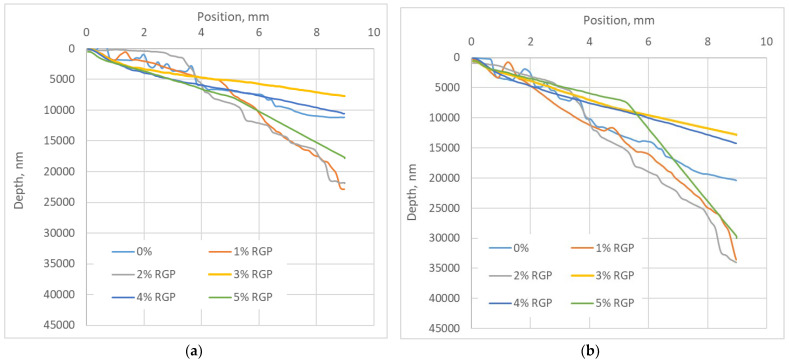
Scratch depth: (**a**) with glass powder at beginning of test; (**b**) with glass powder after 1000 cycles; (**c**) with micro-bubbles at beginning of test; (**d**) with micro-bubbles after 1000 cycles.

**Figure 11 materials-17-03764-f011:**
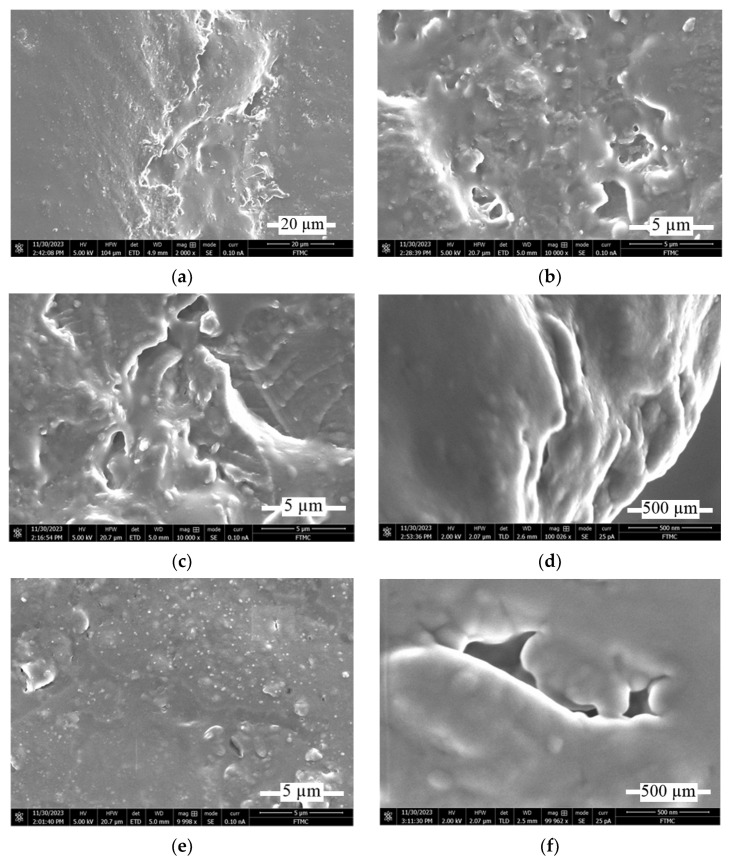
SEM pictures of scratch path of RGP (3% concentration) and RMB (1% concentration) composites at 15 N loading: (**a**) scratch edge of RGP sample at ×2000 magnification; (**b**) scratch center of 3% RGP at ×10,000; (**c**) left from scratch edge of RGP at ×10,000; (**d**) scratch center of RGP at ×100,000; (**e**) scratch path of RMB sample at ×10,000; (**f**) fracture of RMB sample at ×100,000.

**Table 1 materials-17-03764-t001:** Anycubic basic colored UV sensitive resin specification.

Properties	Values	Unit
Viscosity at 25 °C	150–200	mPa·s
Wavelength	405	Nm
Tensile strength	36–45	MPa
Elongation	8–12	%
Flexural strength	50–65	MPa
Flexural modulus	1.2–1.6	GPa
Volume shrinkage	4.5–5.5	%
Heat deflection temperature	65–70	°C
Density	1.05–1.25	g/cm^3^
Hardness (shore D)	82	D
Notched impact strength	25	J/m

**Table 2 materials-17-03764-t002:** Glass powder datasheet.

Properties	Values	Unit
Type	E glass	
Powder shape	Chopped glass fiber	
Powder length dimension	100–200	μm
Moisture content	0.40	%

**Table 3 materials-17-03764-t003:** Printing parameters.

Printing Parameters	Values	Unit
Layer Thickness	0.05	mm
Exposure time	2	s
Off time	1	s
Z lift distance	5	mm
Z lift speed	2	mm/s
Bottom exposure time	35	s
Z retracting speed	2	mm/s

## Data Availability

The original contributions presented in the study are included in the article, further inquiries can be directed to the corresponding authors.

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
