# Peer review of "Tribological Research of Resin Composites with the Fillers of Glass Powder and Micro-Bubbles"

_materials, 2024, doi:10.3390/ma17153764_

Round 1

Reviewer 1 Report

Comments and Suggestions for Authors

The topic of research is very interesting and also the research is done in a true path and in excellent level. In addition, the current study includes valuable results. One of the main advantages of this manuscript is the discussion section and interpretation of the results, which shows the expertise of the authors in this field. However, this manuscript can be published in the present form and need major revision. To this end, the authors should be considered the following items:

1- Introduction section is acceptable and also the novelty is described in short, but it is better to extend the literature review using more published papers along with more details.

2- Regarding to the testing samples, more details should be explained such as the sample size, number of samples, etc. 

3- As I found, the samples are prepared by 3D printing technique. So, it is necessary to report the setting and process parameters, etc. 

4- There are some repeated sentences in the text that should be removed through manuscript. 

5- On page 6 of 12 and line 194, it is stated that the SEM analysis is done on samples run as at 250 rpm, this case is not in the testing plan. the testing plan was for 200 to 900 rpm with interval 100 rpm and also 500 rpm and variable force. 

6- Regarding to all SEM images, i.e., Figures 5, 6, and 9, please notice that details should be shown on images and also, the location of zoom in area should determine in the first image. 

Comments on the Quality of English Language

It is strongly suggested to edit and review the final version of the manuscript by a Native English Person. There is a lot of typo and also some sentences are not clear to understand. 

Author Response

Thank you for the reviewer remarks. In attached file we include the answers to those remarks. The manuscript were revised and all revisions are marked in yellow.

Reviewer 2 Report

Comments and Suggestions for Authors

Padgurskas et al studied the tribological properties of the composites that are based on resins, glass powder and microbubbles. These tribological tests were performed using block on the ring scheme under dry frictional environment. The microscopic tests show the structural damage that include wear losses. Interesting results are reported and the results are nicely presented. However, to improve the paper further. Some important points need to be considered are –

[1] Please summarize the key takeaways shortly in abstract. All provide some potential industrial usefulness of this work.

[2] In last paragraph of introduction, please highlight the novelty of present work more efficiently.  Why this work is important and its advancement over existing literature is desirable.

[3] In materials and methods section, please provide the physical and chemical properties of the ingredients used in this work.

[4] In Figure 4, the probability of wear loss depends upon the increasing speed, increasing load and the filler content. So, what is the main mechanism behind this behavior.

[5] The section 3.3 shows the microscopic studies. The scale bar is not visible. So, please make it bold. Also, what authors want to convince from these micrographs is not clear since the interpretation is highly confusing.

[6] From Figure 7, please provide the elemental composition in form of table as the elements and their respective peaks is not visible.

[7] Please provide the optical pictures of the all the experimental set-ups used in present work.

[8] The conclusion section need to be more informative and must details all the key take away from experiments more clearly.

Author Response

(The authors gave the same response as above.)

Reviewer 3 Report

Comments and Suggestions for Authors

Dear authors,

The research theme approached in your manuscript is interesting concerning to the novel materials obtained by adding fillers to resins in order to improve the tribological behavior.

The topic of this manuscript is indeed very well related with the topics of Materials journal.

In order to improve this interesting manuscript, I would like to recommend some major changes and improvements as is shown below.

1. In the end of Introduction section, the authors must clarify the main purpose of their research and the tests performed on the materials involved in this research in order to reach the main objectives of their research.

2. The chapter 2 should be divided in two sub-sections: 2.1. Materials tested; 2.2. Work method.

3. In sub-section 2.2, the authors should describe in detail all tests carried-out and all equipment used.

4. More details are required for each equipment used: type, manufacturer, some technical details.

5. In section 2.1, the authors must clarify the shape of the specimens tested. According with Figure 1, the shape of the specimens is not parallelepiped shape. Dimensions are 18 mm x 10 mm x 6 mm in the related text.

6. The authors must show a photo of the equipment during the tribological testing. How is the force applied? Is it a distributed or concentrated force?

7. How many specimens are considered for a set of specimens made of the same materials and tested with the same testing parameters?

8. The authors should report the average value and stdev for friction coefficient.

9. The text of the paper contains many uncertainties. The authors must give details in section 2.1 about the manufacturing technology. Which are the parameters used for printing? What does it mean UV curring from the test parameters point of view?

Comments on the Quality of English Language

Dear authors,

Please check carefully the text of your manuscript and improve the expression in English. Please, make correction for the grammatical errors.

Author Response

(The authors gave the same response as above.)

Round 2

Reviewer 1 Report

Comments and Suggestions for Authors

the authors tried to provide revised manuscript based on the reviewers comments and also they answered to all comments one by one. In summary, I have no more comments and it can be published in the present form. 

Author Response

No comments

Reviewer 2 Report

Comments and Suggestions for Authors

The authors have revised the paper nicely and satisfactorily. However, minor revision is still required before acceptance. the comment is-

[1] In the last paragraph of the introduction. please highlight the novelty of this work. For example, what is the new contribution of this work to existing literature and how is this work superior in its literature field? 

Author Response

Thank you for the reviewer remarks. 

The novelty of this research has been added to the last paragraph in the introduction.

This study aims to investigate the possibilities of incorporating glass powder and micro-bubbles modified resin composites for tribological applications in friction joints of machinery. Compared with the literature review process, this study also gives a new technique to manufacture vat-resin without adding a complex additive to enhance the tribological properties. Thus, making the manufacturing process much easier.

The manuscript was revised and all revisions are marked in yellow.

Reviewer 3 Report

Comments and Suggestions for Authors

Dear authors,

I read carefully the improved version of your manuscrid and I found that more details were added to the manuscript according to my suggestions and recomandations.

I think that your manuscript can be recommended for publishing in Materials journal.

Comments on the Quality of English Language

Dear authors,

Moderate corrections for the grammatical errors are required.

Author Response

Thank you for the reviewer remark. 

The manuscript grammar has been extensively checked and revised.

The revisions in manuscript are marked in yellow.